# Genetic Factors Causing Thyroid Dyshormonogenesis as the Major Etiologies for Primary Congenital Hypothyroidism: Clinical and Genetic Characterization of 33 Patients

**DOI:** 10.3390/jcm11247313

**Published:** 2022-12-09

**Authors:** Rui Liu, Jing-Li Tian, Xiao-Ling Huang, Yuan-Zong Song

**Affiliations:** 1Department of Pediatrics, The First Affiliated Hospital, Jinan University, Guangzhou 510630, China; 2Department of Pediatrics, Huizhou No. 2 Women’s and Children’s Healthcare Hospital, Huizhou 516000, China; 3Neonatal Screening Center, Dongguan Maternal and Child Healthcare Hospital, Dongguan 523125, China

**Keywords:** congenital hypothyroidism, genetic variants, thyroid dysgenesis, dyshormonogenesis, neonatal screening

## Abstract

Background and aims: Although the significance of primary congenital hypothyroidism (CH) is supported by an increasing amount of evidence, the clinical and genetic characteristics of this condition are still poorly understood. This study aimed to explore the underlying genetic etiologies in a cohort of primary CH patients. Subjects and Methods: The clinical data of 33 patients with primary CH were collected and analyzed via a cross-sectional study. Genetic analysis was performed by high-throughput sequencing and Sanger verification, and the pathogenicity of the novel missense variants was predicted using a variety of comprehensive bioinformatic tools. Results: Among the 33 patients, 22 (22/33, 66.7%) harbored pathogenic variants in the causative genes of thyroid dysgenesis or dyshormonogenesis, with *DUOX2* (15/33, 45.5%) topping the list, followed by *TG*, *TPO*, *DUOXA2* and *PAX8*. Four novel genetic variants were detected, including a pathogenic frameshift and three likely pathogenic missense variants. Positive neonatal screening for TSH, neonatal jaundice and abnormal thyroid morphology were the main positive findings among all cases. Although 31 of the total 33 CH patients exhibited normal anthropometric and social performance, the other 2 had poor prognosis in this study. Conclusions: This study reported 33 new CH patients bearing four novel genetic variants, which enriched the variant spectrum of CH genes. In this cohort, genetic factors causing thyroid dyshormonogenesis were the main etiologies of CH development. Most patients exhibited a favorable prognosis; however, systematic management remains a challenge in achieving improved clinical outcomes for CH patients.

## 1. Introduction

Congenital hypothyroidism (CH) is estimated to occur in approximately 1 in 3000–4000 live births worldwide, and timely diagnosis is crucial, considering the essential role of thyroid hormones in central nervous system maturation [1,2,3]. For years, the most prevalent cause of CH was iodine deficiency, but today, the significance of defects in the thyroid gland itself, which is defined as primary CH, is supported by an increasing amount of evidence [4]. Due to the advent of next-generation sequencing (NGS), more than 24 genes have been identified to be linked with primary CH [5,6], which is divided into thyroid dysgenesis (TD) and thyroid dyshormonogenesis (TDH) [7]. The former, which exhibits abnormal thyroid gland organogenesis, is associated with the genes *TSHR*, *TUBB1*, *PAX8*, *NKX2-1*, *NKX2-5*, *GLIS3 CDC8A* and *NTN1* [3,7,8,9], while the latter, which is characterized by defects in a structurally normal thyroid gland to produce normal quantities of thyroid hormone, is suspected to be caused by the genes *TG*, *TPO*, *SLC26A4*, *SLC26A7*, *SLC5A5*, *DUOX1*, *DUOX2* or *DUOXA2* [3,10,11]. The inheritance model of TDH is generally autosomal recessive; however, the inheritance mode of TD remains debated, with several mechanisms that need to be considered, such as monogenic, polygenic and epigenetic inheritance [12].

The typical presentations of CH include myxedematous face, macroglossia, umbilical hernia, hypotonia, large fontanels, a distended abdomen, decreased activity and increased sleep, constipation, feeding difficulty and prolonged neonatal jaundice [5,13]. However, these clinical manifestations are often subtle at birth, and even negative in some cases with elevated thyroid-stimulating hormone (TSH) levels but normal thyroid hormone levels, making diagnosis of etiology and genetic counseling greatly challenging [14]. In recent years, next-generation sequencing (NGS) has facilitated the identification of the etiology of genetic conditions, and it is now possible to analyze the genetic etiologies of CH patients simultaneously and systematically [7,15].

The incidence of CH varies by race and ethnicity. It has been reported at 1:1016 in Asian people, 1:1815 in Caucasian people and 1:1902 in Black people [16]. China is a vast country with a huge population, and a TSH-based neonatal screening for CH revealed an incidence of approximately 1 in 2000–2500 live births [10,17,18]. However, the clinical and genetic characterization of this condition remain poorly understood in our country. The present study aimed to explore the underlying genetic etiologies in a cohort of primary CH patients.

## 2. Materials and Methods

### 2.1. Subjects and Ethical Approval

A total of 33 patients from November 2006 to July 2021, including three pairs of siblings (patients 1 and 2, 3 and 4 and 11 and 12), with primary CH were recruited as the research subjects. We recruited patients via a TSH-based neonatal screening program taking place between 72 h and 7 days after their births, with blood samples collected from the heel, and the TSH levels were measured by time-resolved fluorescence assay. All diagnoses were confirmed by at least one recall test(s) of the serum TSH, free thyroxine (FT3) and FT4, which were analyzed by electrochemistry immunoassay. CH was diagnosed based on TSH elevation (TSH ≥ 10 mIU/L) with low FT4 levels (FT4, with the reference range 12–27 pmol/L), and thereafter treated with levothyroxine (L-T4) at an initial dosage of 10–15 μg/kg per day, which was adjusted according to the serum levels of TSH and FT4 [14,19,20,21,22]. Patients with known causes of transient CH, including maternal use of antithyroid drugs, premature birth gestation (<37 weeks), history of iodine deficiency or iodine excess during pregnancy and patients who refused to undergo genetic testing or who were lost to follow-up were excluded. All the research subjects in this study met the CH diagnostic criteria [19,20,21].

The current study was conducted in accordance with the Declaration of Helsinki, and approved by the Committee for Medical Ethics, The First Affiliated Hospital of Jinan University (KY-2019-048 approved on 28 November 2019 and KY-2021-001 on 18 January 2021) and written informed consent was obtained from the parents of the patients.

### 2.2. Clinical Evaluation

Clinical data including main complaints, history, physical and laboratory examination, treatment and outcomes were collected via a comprehensive review of the patients’ medical records on serial clinic follow-up. Notably, all CH patients underwent a detailed thyroid ultrasonography examination in the longitudinal and axial planes by thyroid imaging specialists who had at least 4 years of experience in pediatric ultrasonography. Moreover, developmental quotient (DQ) evaluation was performed during the period of treatment by board-certified physiatrists. The scales used for DQ evaluation included Griffiths Mental Development Scales (GMDSs) and Denver Developmental Screening Tests (DDSTs).

### 2.3. Targeted NGS Analysis

Genomic DNA was extracted from peripheral blood leukocytes using a DNA extraction kit (Simen) according to the manufacturer’s protocol to create customized NimbleGen SeqCap probes (Roche NimbleGen, Madison, WI, USA) containing target genes associated with thyroid diseases, such as *DUOX2*, *DUOXA2*, *TG*, *TSHR*, *TPO* and *PAX8.* All exons and the flanking nucleotide sequences of the targeted genes were captured using a liquid-phase capture kit (Mygenostics, Beijing, China), and NGS was performed on the Illumina Hiseq 2000 sequencer (Illumina, San Diego, CA, USA). The overall sequencing coverage of the target regions was >95%, and the average depth of sequencing was not less than 200×. Sequence variant nomenclature was in agreement with current guidelines of the Human Genome Variation Society (http://www.hgvs.org/rec.html, accessed on 26 July 2022).

### 2.4. Sanger Sequencing

Sanger sequencing was performed to validate the relevant variants detected on NGS. DNA fragments were amplified by polymerase chain reaction (PCR) in 100μL of reaction system with the following temperature conditions: pre-denaturation at 98 °C for 1 min; 35 cycles of denaturation at 98 °C for 25 s, annealing at 65 °C for 30 s and elongation at 72 °C for 30 s; and a final extension at 72 °C for 5 min. The variants were identified by comparison with the reference sequence on the National Center Biotechnology Information (NCBI) website (https://www.ncbi.nlm.nih.gov/,accessed on 26 July 2022).

The sequencing results were aligned with the known gene sequence, which is available at Ensembl Genome Browser (www.ensembl.org, accessed on 26 July 2022), using the DNAman software (version 5.2.2; Lynnon Biosoft Corporation, San Ramon, CA, USA) and analyzed using the Chromas software (version 2.6.6; Technelysium Pty, Ltd., South Brisbane, Australia). The allele frequency of the identified novel variant was investigated in five population databases, including the 1000 Genomes Project (https://www.internationalgenome.org/, accessed on 26 July 2022), the Genome Aggregation Database (gnomAD, http://www.gnomad-sg.org/, accessed on 26 July 2022), the Exome Sequencing Project (https://esp.gs.washington.edu/drupal/, accessed on 26 July 2022), the Exome Aggregation Consortium (https://exac.broadinstitute.org, accessed on 26 July 2022) and the Human Gene Mutation Database (https://www.hgmd.cf.ac.uk/ac/index.php, accessed on 26 July 2022). Variants were filtered for allele frequencies < 1%. All the variants were classified according to the American College of Medical Genetics and Genomics (ACMG) guidelines [23].

### 2.5. Bioinformatic Analysis

A total of five prediction programs were used to predict the pathogenicity of the novel missense variants. PolyPhen-2 (http://genetics.bwh.harvard.edu/pph2/, accessed on 26 July 2022) analysis identifies a variant as ‘probably damaging’ if the probability is >0.85 and ‘possibly damaging’ if the probability is >0.15 [24]. SIFT (http://sift.jcvi.org/www/SIFT_chr_coords_submit.html, accessed on 26 July 2022) identifies the variant as being ‘deleterious’ if the SIFT score is <0.05 [25]. MutationTaster (http://mutationtaster.org/MutationTaster/index.html, accessed on 26 July 2022) prediction identifies probabilities ~1 as a ‘high security’ of the prediction [26]. PROVEN (http://provean.jcvi.org/index.php, accessed on 26 July 2022) analysis identifies a variant as ‘deleterious’ if the score is equal to or less than −2.5, and ‘neutral’ if the score is above −2.5 [27]. MutationAssessor (http://mutationassessor.org/, accessed on 26 July 2022) scores a mutation by global and subfamily specific conservation patterns as low, medium or high [28].

Comparative alignment of the amino acid sequences of *DUOX2*, *TG* and *PAX8* orthologs, collected from 151, 114 and 170 species, respectively, from the Ensembl Genome Browse (https://asia.ensembl.org/index.html, accessed on 26 July 2022), was performed. The 151 *DUOX2* orthologs were classified into five taxonomy subgroups: primates (24 species), rodents and lagomorphs (33 species), other mammals (47 species), other vertebrates (44 species) and other species (3 species). The 114 *TG* orthologs were also classified into five taxonomy subgroups: primates (22 species), rodents and lagomorphs (30 species), other mammals (27 species), other vertebrates (33 species) and other species (2 species). Meanwhile, the 170 *PAX8* orthologs were classified into four taxonomy subgroups: primates (21 species), rodents and lagomorphs (37 species), other mammals (45 species), other vertebrates (67 species) and other species (0 species).

Moreover, the three-dimensional (3D) structural models of DUOX2, TG and PAX8 protein were built using the online software SWISS-MODEL automated protein modeling server (https://swissmodel.expasy.org/, accessed on 26 July 2022). The tertiary structures of the wild-type and mutant protein were compared using SWISSPdb Viewer 4.1.0 (http://www.expasy.org/spdbv, accessed on 26 July 2022) to evaluate the effect of the variant on the structure of the molecule.

## 3. Results

### 3.1. Results of Genetic Testing

Among the 33 CH patients studied, 22 were found to have genetic causes, with a total rate of 66.7% (22/33). The genes causing TD (*PAX8*) and TDH (*DUOX2*, *DUOXA2*, *TG* and *TPO*) accounted for 3.0% (1/33) and 63.6% (21/33) of total cases, respectively (Table 1, Figure 1). *DUOX2* was the most common CH-causing gene (15/33, 45.5%), followed by *TG* (2/33, 6.1%), *TPO* (2/33, 6.1%), *DUOXA2* (2/33, 6.1%) and *PAX8* (1/33, 3.0%). The genetic etiologies remained unclear in the other 11 (11/33, 33.3%) CH patients (Appendix A).

Of note, c.3904T > C and c.2018dupA in *DUOX2* as well as c.7404G > C in *TG* were detected in trans with another reported pathogenic mutation, besides a heterozygous variant, c.307T > A, in the *PAX8* gene, in the relevant genes (Figure 2). To the best of our knowledge, these four variants, including three missenses and one duplication, have not been previously reported in any official references in the database Pubmed; however, the Exome Aggregation Consortium database has recorded the variant c.7404G > C with an allele frequency of 0.004‰, and gnomAD cited the variants c.3904T > C and c.2018dupA with an allele frequency of 0.00398‰ and 0.00398‰. The four novel variants are not currently included in the 1000 Genomes Project, the Exome Sequencing Project or the Human Gene Mutation Database.

### 3.2. Bioinformatic Findings

In ployphen-2 analysis, the missense variants c.3904T > C (p.Trp1302Arg) and c.7404G > C (p.Lys2468Asn) were predicted to be ‘probably damaging’, while the c.307T > A (p.Phe103Ile) variant was predicated to be ‘benign’. The three variants were all predicted to be ‘disease-causing’ on Mutation Taster and ‘deleterious’ on PROVEAN and SIFT. Moreover, MutationAssessor yielded a result of ‘high’ for the first variant, ‘medium’ for the second and ‘neutral’ for the last. The results are shown in Appendix A.

To predict the pathogenicity of the three novel missense variants, the affected amino acids in a total of 151, 114 and 170 peptides homologous to human *DUOX2*, *TG* and *PAX8*, respectively, were comparatively aligned. The amino acids p.Trp1302 in *DUOX2* and p.Phe103 in *PAX8* were consistent in all homologous peptides (Appendix A). The amino acid p.Lys2468 in *TG* was relatively conserved in 22/22 of the primates including humans (Appendix A), 30/30 of rodents and lagomorphs, 26/27 of the other mammals, 25/33 of the other vertebrates and 2/2 of the other species.

Moreover, in SWISS-MODEL analysis (Appendix A), the three novel missense variants exhibited alterations in the length and/or number of the hydrogen bonds between specific amino acid residues in the DUOX2, TG and PAX8 proteins, thus distorting the molecular structure of the relevant protein.

### 3.3. Clinical Presentations

Among the 33 CH patients, 16 (16/33, 48.5%) presented with neonatal jaundice, 2 (2/33, 6.1%) with increased thyroid gland sizes and 4 (4/33, 12.1%) with thyroid aplasia, while another 2 (2/33, 6.1%) presented with thyroid ectopy in ultrasonography analysis (Table 2).

Oral L-T4 was given to 29 (29/33, 87.9%) CH patients as early as the neonatal period. However, in the other four (4/33, 12.1%) cases (patients 11, 21, 23 and 25), L-T4 supplementation was initiated at ages 36, 36, 46 and 90 days, respectively, due to incompliance or inaccessibility to a timely recall test, treatment and regular clinic follow-up (Table 3 and Appendix A).

As a result of L-T4 uptake, the thyroid function indices in most cases (31/33, 93.9%) generally improved within eight weeks (Table 3 and Appendix A). Although elevated TSH levels were observed occasionally in 24 patients (24/33, 72.7%), they all ultimately improved or recovered in response to subsequent adjustment of the L-T4 dosage (Appendix A). Then, the L-T4 dosages were gradually reduced, and clinic follow-up was performed in 29 CH patients. Of note, in the other four affected individuals (patients 12, 13, 29 and 31), L-T4 treatment was eventually stopped at ages 4.5, 3.5, 0.1 and 4.3 years and thereafter their TSH levels were observed to be normal.

Regarding the clinical outcomes (n = 33) at the latest follow-up, at the age of 47 (17.5, 71.5) months, 28 patients exhibited normal anthropometric and social performance with a normal developmental quotient (DQ) (Appendix A). Although DQ results are not available for patients 3, 4 and 26, no abnormal anthropometric or social performance was observed by their parents or school teachers. However, patient 21 received a low DQ on evaluation with retardation of personal–social skills, fine motor skills and language development, while patient 25 exhibited microcephaly along with a significantly poor learning ability and school performance due to incompliance/inaccessibility to the recall test, treatment and regular follow-up (Table 2 and Appendix A).

## 4. Discussion

This study reported 33 new primary CH patients and identified four novel genetic variants. The molecular findings suggest that genetic factors constituted the main (61.1%, 22/33) CH etiologies in this cohort. According to the ACMG standards [23], the novel *DUOX2* variant c.2018dupA (p.Val674Glyfs*39) was classified as pathogenic; since frameshift disrupted gene function (PVS1), the novel variant appeared in trans with known pathogenic variants (PM3), and the clinical phenotypes of the patient were specific for CH (PP4). Additionally, the novel *DUOX2* variant c.3904T > C (p.Trp1302Arg), *TG* variant c.7404G > C (p.Lys2468Asn) and *PAX8* variant c.307T > A (p.Phe103Ile) were all classified as likely pathogenic. In detail, they were absent or found at extremely low frequencies in the 1000 Genomes Project, Exome Sequencing Project, gnomAD and Exome Aggregation Consortium (PM2). The novel variants appeared in trans with known pathogenic variants (PM3), and a variety of bioinformatic tools suggested that the three novel missense variants may be detrimental to the functionality of the protein (PP3). Finally, the clinical phenotype of the four patients were quite specific for CH (PP4). It was well known that, when combined with other evidence of the disease in question, a variant classified as likely pathogenic has sufficient evidence for a health-care provider to use the molecular testing information in clinical decision making [23]. In combination with the clinical and laboratory findings, the four novel genetic variants enriched the variant spectrum of CH-associated genes and provided important molecular evidence for the diagnosis of etiology and genetic counseling for CH patients and their families.

The *DUOX2*- and *TG*-encoded proteins play important roles in the process of thyroid hormonogenesis. *DUOX2* encodes a dual oxidase to generate hydrogen peroxide, which is needed by thyroid peroxidase for the incorporation of iodine into thyroglobulin [29]. The amino acid residue p.Trp1302 is located in the flavine adenosine dinucleotide (FAD)-binding domain of the DUOX2 protein (https://www.uniprot.org/, accessed on 26 July 2022), and the novel *DUOX2* variant c.3904T > C (Trp1302Arg) distorted the domain structure and hence might impair its FAD-binding ability. Another novel *DUOX2* variant c.2018dupA gives rise to a truncated protein (p.Val674Glyfs*39) lacking the C-terminal FAD-binding domain and two EF-hand motifs (https://www.uniprot.org/, accessed on 26 July 2022), which inevitably affect the dual oxidase activity of DUOX2 needed to generate hydrogen peroxide. Additionally, *TG*-encoded thyroglobulin functions as the substrate for the synthesis of thyroid hormones [5]. The novel *TG* variant c.7404G > C (p.Lys2468Asn) affects the Cholinesterase-like (ChEL) domain of the TG protein (https://www.uniprot.org/, accessed on 26 July 2022). Since a variety of ChEL domain mutants impair upstream TG folding and eliminate TG secretion from the apical membrane of the follicular cells to iodination sites in the follicle [30,31,32], this variant might cause CH via a similar mechanism by changing the TG molecule structure.

The gene *PAX8* encodes the PAX8 protein, which is involved in the development of thyroid follicular cells and plays an essential role in the regulation of thyroid development, functional differentiation and organogenesis in human embryos [33,34], acting as a transcription factor characterized by the presence of a DNA-binding paired domain at the N-terminus [35]. The *PAX8* novel variant c.307T > A affects the amino acid residue p.Phe103, which corresponds to its DNA-binding domain (https://www.uniprot.org/, accessed on 26 July 2022) and causes structural distortion of the PAX8 molecule via changing hydrogen bonds (Appendix A). It has been reported that *PAX8* variants involved in this domain usually cause reduced *TG* and *TPO* promoter transcriptional activation ability in transient transfection studies, due to an impaired DNA-binding ability; thus, the differentiation of thyroid cells could be influenced [33,34]. Overall, it is not surprising that the individuals harboring these four novel genetic variants presented with positive neonatal screening for TSH, neonatal jaundice and/or abnormal thyroid morphology, as shown in Table 2.

In the 22 CH patients with positively detected genetic causes, only one patient was associated with the TD-related gene PAX8, while 21 were associated with the TDH-related genes *DUOX2*, *DUOXA2*, *TG* and *TPO* (Table 1), suggesting that TDH is more common than TD as the major mechanism for CH development. This finding is consistent with previous publications [6,9,36,37]. Among the TDH-associated genes positively detected in this CH cohort, as shown in Table 1 and Figure 1, *DUOX2* topped the list, followed by *DUOXA2*, *TG* and *TPO*. Of note, *DUOX2* variants were frequently detected in CH patients of Asian origin, including those from China, Japan and Korea [10,36,38,39]. The mutated *DUOX2* allele frequencies in the primary CH patients from the Chinese cities of Changzhou, Shanghai, Guangxi, Qingdao and Guangzhou were as high as 37.1%, 32.8%, 29.0%, 21.2% and 31.3%, respectively [40,41,42,43,44]. On the other hand, TD has been reported as the most common cause of CH [5,16,45,46], and the involved genes include *TSHR*, *PAX8*, *NKX2-1*, *NKX2-5*, *GLIS3* and *NTN1* [3,7,8,9]. Considering the possible sampling bias in this study, data for a larger sampling size are still in need to evaluate the significance of TDH and TD in CH development.

Additionally, there were still 11 patients with typical CH manifestations but unclear underlying genetic etiologies. This finding might have arisen to due to the following reasons. First, these patients might carry hidden genetic variants such as large deletions/insertions, variants in the noncoding regions affecting transcription factors or copy number variations (CNVs) [47]. Second, there might be more relevant CH-causing genes which have not been discovered yet [5]. Third, other modes of inheritance including epigenetics and multigenic involvement could not be excluded [9]. Actually, patient 27 in this study had oligogenic variants in the genes *DUOX2* and *TG*, and the possibility of an oligogenic mechanism, a specific multigenic manner, for his CH development could not be completely ruled out.

It is well-known that oral L-T4 is the main treatment for primary CH patients. However, this treatment does not need to continue for the whole life duration of patients, especially in those with transient but not permanent CH [6,14,41,48]. The penetrance was 100% in the 22 CH-affected children in this study, and even in such cases, L-T4 withdrawal was achieved in patients 12, 13, 29 and 31 (Table 3). This phenomenon might be explained by the fact that the demand for thyroid hormone is five to seven times higher in the neonatal period than in the adult stage, and gradually reduces after the infantile period [41]. As a possible explanation, functional compensation might exist in CH-affected subjects. For example, patient 12 in this study exhibited biallelic *DUOX2* variants (Table 1), and *DUOX1* can compensate for the *DUOX2* mutation in thyroid cells to produce H_2_O_2_. The insufficient supply of H_2_O_2_ using *DUOX1* alone in the neonatal and infantile periods might become sufficient as thyroid hormone requirements decrease with age [41,48].

Once diagnosed and treated in time, primary CH patients usually demonstrate benign prognosis. Unfortunately, due to incompliance or inaccessibility to a timely recall test, treatment and regular clinic follow-up, two patients in this study exhibited intelligence impairment, and similar unfavorable prognoses have been observed in previous studies [42,48,49,50]. These findings strongly indicate that, although general neonatal screening has been performed for years, the systematic management of CH patients by timely recall tests, diagnosis, treatment and regular clinic follow-up remains an issue today.

This study reported 33 new CH patients bearing four genetic variants, which enriched the variant spectrum of CH genes, and provided genetic evidence supporting the diagnosis of etiology and genetic counseling for the affected individuals. In this cohort, genetic factors causing thyroid dyshormonogenesis were the main etiologies of CH development. Most patients exhibited favorable clinical outcomes, and in some transient cases, the L-T4 treatment was temporary. However, the systematic management of this condition remains a significant challenge.

## Figures and Tables

**Figure 1 jcm-11-07313-f001:**
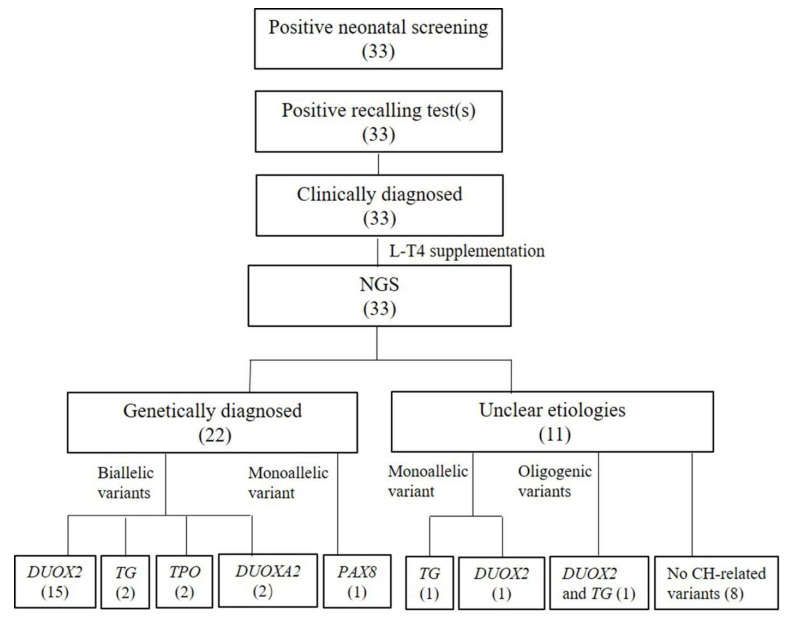
Genetic analytic findings in the 33 CH patients.

**Figure 2 jcm-11-07313-f002:**
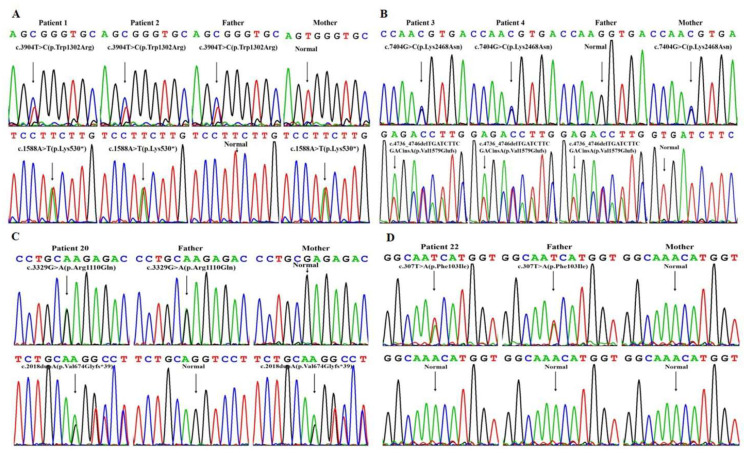
Sanger sequencing results of *DUOX2*, *TG* and *PAX8* gene in Patients 1, 2, 3, 4, 20 and 22. (**A**) Patients 1 and 2 were compound heterozygous for the *DUOX2* variants c.3904T > C (p.Trp1302Arg) and c.1588A > T (p.Lys530*), while their father was a carrier of c.3904T > C and their mother was a carrier of c.1588A > T. (**B**) Patients 3 and 4 were compound heterozygous for the *TG* variants c.4736_4746delTGATCTTCGACinsA (p.Val1579Glufs) and c.7404G > C (p.Lys2468Asn) from their father and mother, respectively. (**C**) Patient 20 was compound heterozygous for the *DUOX2* variants c.3329G > A (p.Arg1110Gln) and c.2018dupA (p.Val674Glyfs*39) from father and mother, respectively. (**D**) Patient 22 was a CH patient with *PAX8* variant c.307T > A (p.Phe103Ile) from her father, while her mother was normal.

**Table 1 jcm-11-07313-t001:** Genotyping results of the 22 CH patients with clear genetic etiologies.

Patients	MOI	Causative Genes	The Allele Frequency in gnomAD	Genotypes		
				Index Patient	Father	Mother
1	AR	* DUOX2 *	0.00398‰/0.0119‰	c.3904T > C (p.Trp1302Arg)/c.1588A > T (p.Lys530*)	c.3904T > C (p.Trp1302Arg)/-	c.1588A > T (p.Lys530*)/-
2	AR	* DUOX2 *	0.00398‰/0.0119‰	c.3904T > C (p.Trp1302Arg)/c.1588A > T (p.Lys530*)	c.3904T > C(p.Trp1302Arg)/-	c.1588A > T (p.Lys530*)/-
3	AR	* TG *	NA/0.00398‰	c.4736_4746delTGATCTTCGACinsA (p.Val1579Glufs)/c.7404G > C (p.Lys2468Asn)	c.4736_4746delTGATCTTCGACinsA (p.Val1579Glufs)/-	c.7404G > C (p.Lys2468Asn)/-
4	AR	* TG *	NA/0.00398‰	c.4736_4746delTGATCTTCGACinsA (p.Val1579Glufs)/c.7404G > C (p.Lys2468Asn)	c.4736_4746delTGATCTTCGACinsA (p.Val1579Glufs)/-	c.7404G > C (p.Lys2468Asn)/-
5	AR	* TPO *	0.117‰	c.2268dupT (p.Glu757*)/c c.2268dupT (p.Glu757*)	c.2268dupT (p.Glu757*)/-	c.2268dupT (p.Glu757*)/-
6	AR	* DUOX2 *	0.598‰; 0.343‰/ 0.417‰	[c.4027C > T (p.Leu1343Phe); c.2048G > T (p.Arg683Leu)]/ c.2654G > T (p.Arg885Leu)	[c.4027C > T (p.Leu1343Phe); c.2048G > T (p.Arg683Leu)]/-	c.2654G > T (p.Arg885Leu)/-
7	AR	* DUOX2 *	0.417‰/0.113‰	c.2654G > T (p.Arg885Leu)/c.2654G > A (p.Arg885Gln)	c.2654G > T (p.Arg885Leu)/-	c.2654G > A (p.Arg885Gln)/-
8	AR	* DUOX2 *	0.343‰; 0.598‰/ 0.191‰; NA	[c.2048G > T (p.Arg683Leu); c.4027C > T (p.Leu1343Phe/ [c.3329G > A (p.Arg1110Gln); c.1094A > G (p.Gln365Arg)]	[c.2048G > T (p.Arg683Leu); c.4027C > T (p.Leu1343Phe)]/-	[c.3329G > A (p.Arg1110Gln); c.1094A > G (p.Gln365Arg)]/-
9	AR	* DUOX2 *	0.0119‰/ 0.598‰; 0.343‰	c.1588A > T (p.Lys530*)/ [c.4027C > T (p.Leu1343Phe); c.2048G > T (p.Arg683Leu)]	c.1588A > T (p.Lys530*)/-	[c.4027C > T (p.Leu1343Phe); c.2048G > T (p.Arg683Leu)]/-
10	AR	* DUOX2 *	0.191‰/0.0191‰	c.3329G > A (p.Arg1110Gln)/c.1588A > T (p.Lys530*)	c.3329G > A (p.Arg1110Gln)/-	c.1588A > T (p.Lys530*)/-
11	AR	* DUOX2 *	0.0119‰/ 0.343‰; 0.598‰	c.1588A > T (p.Lys530*)/ [c.2048G > T (p.Arg683Leu); c.4027C > T (p.Leu1343Phe)]	c.1588A > T (p.Lys530*)/-	[c.2048G > T (p.Arg683Leu); c.4027C > T (p.Leu1343Phe)]/-
12	AR	* DUOX2 *	0.019‰/ 0.343‰; 0.598‰	c.1588A > T (p.Lys530*)/ [c.2048G > T (p.Arg683Leu); c.4027C > T (p.Leu1343Phe)]	c.1588A > T (p.Lys530*)/-	[c.2048G > T (p.Arg683Leu); c.4027C > T (p.Leu1343Phe)]/-
13	AR	* DUOXA2 *	0.231‰/0.143‰	c.413dupA (p.Tyr138*)/c.738C > G (p.Tyr246*)	c.413dupA (p.Tyr138*)/-	c.738C > G (p.Tyr246*)/-
14	AR	* DUOX2 *	0.417‰/0.113‰	c.2654G > T (p.Arg885Leu)/c.3693 + 1G > T	c.2654G > T (p.Arg885Leu)/-	c.3693 + 1G > T/-
15	AR	* DUOXA2 *	0.231‰/0.00402‰	c.413dupA (p.Tyr138*)/c.515dupA (p. Tyr173Valfs*57)	c.413dupA (p.Tyr138*)/-	c.515dupA (p. Tyr173Valfs*57)/-
16	AR	* DUOX2 *	0.0707‰/0.113‰	c.2104_2106delGGA (p.Gly702del)/c.3693 + 1G > T	c.2104_2106delGGA (p.Gly702del)/-	c.3693 + 1G > T/-
17	AR	* DUOX2 *	0.0119‰/0.0679‰	c.1588A > T(p.Lys530*)/c.1946C > A(p.Ala649Glu)	c.1588A > T(p.Lys530*)/-	c.1946C > A (p.Ala649Glu)/-
18	AR	* DUOX2 *	0.191‰/0.0119‰	c.3329G > A(p.Arg1110Gln)/c.1588A > T(p.Lys530*)	c.3329G > A(p.Arg1110Gln)/-	c.1588A > T (p.Lys530*)/-
19	AR	* DUOX2 *	0.0119‰/0.0278‰	c.1588A > T (p.Lys530*)/c.3478_3480del (p.Leu1160del)	c.1588A > T (p.Lys530*)/-	c.3478_3480del (p.Leu1160del)/-
20	AR	* DUOX2 *	0.191‰/0.00398‰	c.3329G > A (p.Arg1110Gln)/c.2018dupA (p.Val674Glyfs*39)	c.3329G > A (p.Arg1110Gln)/-	c.2018dupA (p.Val674Glyfs*39)/-
21	AR	* TPO *	0.117‰/NA	c.2268dupT (p.Glu757*)/c.1804C > T (p.Arg602Cys)	c.2268dupT (p.Glu757*)/-	c.1804C > T (p.Arg602Cys)/-
22	AD	* PAX8 *	NA/-	c.307T > A (p.Phe103Ile)/-	c.307T > A (p.Phe103Ile)/-	-/-

MOI: Mode of inheritance; AD: Autosomal dominant; AR: Autosomal recessive; *TPO*: Thyroid peroxidase; *TG*: Thyroglobulin; *DUOX2*: Dual oxidase 2; *DUOXA2*: dual oxidase maturation factor 2; *PAX8*: Paired Box Gene 8 Mutations; NA: Frequency of the variant was not available in gnomAD.

**Table 2 jcm-11-07313-t002:** Clinical information of the 33 CH patients.

Patients	Sex	GA(Weeks)	BW(g)	Delivery Mode	Chief Complaint	Family History	ThyroidMorphology/Age at Latest Examination (y)	OtherPresentations	Clinical Outcomes/Assessment Ages (y)
1	Female	40^+2^	2700	Vaginal delivery	Abnormal NS	N	Normal/3.0	None	Normal DQ/3.0
2	Male	39^+3^	2580	Vaginal delivery	Abnormal NS	His sister had CH	Normal/0.1	None	Normal DQ/0.5
3	Female	41	4500	Caesarean	Abnormal NS	N	Increased/9.3	Macrosomia	Normal school performance/10.0
4	Male	40	3500	Caesarean	Abnormal NS	His sister had CH	Normal/2.7	None	Normal school performance/7.3
5	Male	40^+4^	2900	Vaginal delivery	Abnormal NS	Mother hadHashimoto’s thyroiditis	Normal/3.3	None	Normal DQ/6.7
6	Male	40	3800	Caesarean	Abnormal NS	N	Normal/3.8	None	Normal DQ/3.7
7	Male	38	2450	Vaginal delivery	Abnormal NS	N	Normal/2.3	Neonatal jaundice	Normal DQ/4.3
8	Male	40^+1^	3000	Vaginal delivery	Abnormal NS	N	Normal/0.1	Neonatal jaundice	Normal DQ/0.8
9	Female	42^+3^	3000	Caesarean	Abnormal NS	N	Normal/0.1	None	Normal DQ/0.9
10	Female	39^+1^	3300	Caesarean	Abnormal NS	N	Increased/0.1	Neonatal jaundice	Normal DQ/2.0
11	Female	38	2400	Vaginal delivery	Abnormal NS	N	Normal/3.7	Neonatal jaundice	Normal DQ/4.4
12	Male	40	3250	Vaginal delivery	Abnormal NS	His sister had CH	Normal/2.6	None	Normal DQ/6.3
13	Female	38	2970	Vaginal delivery	Abnormal NS	Her brother had CH	Normal/9.2	None	Normal DQ/4.8
14	Male	37^+5^	2590	Caesarean	Abnormal NS	N	Normal/2.5	One of the monozygotic twins, Full term	Normal DQ/3.9
15	Male	38^+5^	3150	Vaginal delivery	Abnormal NS	N	Normal/1.5	Neonatal jaundice	Normal DQ/3.0
16	Female	39	3500	Caesarean	Abnormal NS	Mother hadHashimoto’s thyroiditis	Normal/3.5	Neonatal jaundice	Normal DQ/4.4
17	Female	39^+1^	3400	Vaginal delivery	Abnormal NS	N	Normal/0.3	Neonatal jaundice	Normal DQ/4.0
18	Male	38	2500	Caesarean	Abnormal NS	N	Normal/3.5	None	Normal DQ/3.5
19	Female	39	3400	Vaginal delivery	Abnormal NS	N	Normal/0.1	None	Normal DQ/0.8
20	Female	39^+1^	3100	Vaginal delivery	Abnormal NS	Her sister had CH	Normal/0.1	Neonatal jaundice	Normal DQ/1.2
21	Male	41	3500	Vaginal delivery	Abnormal NS	N	Normal/0.7	Neonatal jaundice	Low DQ/4.5
22	Female	37	2740	Vaginal delivery	Abnormal NS	Her father had a history of hypothyroidism	Normal/0.1	None	Normal DQ/2.2
23	Female	39^+4^	2780	Vaginal delivery	Abnormal NS	N	Not done	Neonatal jaundiceAtrial septal defect, Pierre-Robin syndrome	Normal DQ/0.9
24	Female	40^+4^	3000	Vaginal delivery	Abnormal NS	Aunt had hyperthyroidism	Thyroid agenesis/0.1	None	Normal DQ/2.5
25	Male	39^+6^	2700	Vaginal delivery	Abnormal NS	Mother had hyperthyroidism in late pregnancy	Not done	None	Microcephaly and poor learning ability and school performance/6.3
26	Female	37^+3^	900	Caesarean	Abnormal NS	N	Not done	One of the monozygotic twins, Full term, Small for gestational age	Normal developmental milestones/0.8
27	Female	39	3100	Vaginal delivery	Abnormal NS	N	Thyroid agenesis/0.1	Neonatal jaundice	Normal DQ/6.1
28	Female	40^+6^	2900	Vaginal delivery	Abnormal NS	N	Thyroid ectopy/4.7	Neonatal jaundice	Normal DQ/3.0
29	Male	38	3450	Caesarean	Abnormal NS	Mother had hyperthyroidism in pregnancy	Normal/0.1	Neonatal jaundice	Normal DQ/0.3
30	Female	38^+5^	3330	Caesarean	Abnormal NS	N	Thyroid ectopy/0.1	None	Normal DQ/2.8
31	Male	40	3370	Vaginal delivery	Abnormal NS	Mother had hypothyroidism	Normal/0.8	Neonatal jaundice	Normal DQ/2.6
32	Female	41	3350	Caesarean	Abnormal NS	Mother had hyperthyroidism	Thyroid agenesis/2.4	Neonatal jaundice	Normal DQ/1.6
33	Female	41	3870	Caesarean	Abnormal NS	N	Thyroid agenesis/1.1	Neonatal jaundice	Normal DQ/1.1

NS: newborn screening; N: no family history of thyroid diseases; GA: gestational age; BW: Birth weight; y: years; CH: congenital hypothyroidism; DQ: Developmental quotient.

**Table 3 jcm-11-07313-t003:** Thyroid function indices over time in the 33 CH patients.

Patients	Age When L-T4 Initiated (days)	Diagnostic Evaluation					Initial L-T4 Dosage (µg/kg/day)	Days of Age at Normalization of Thyroid Function (Days)	Maintenance L-T4 Dosage (µg/kg/day)/ the Latest Ages (Years)
		FT3 (3.5–10 pmol/L)	FT4 (12–27 pmol/L)	T3 (1.3–3.8 nmol/L)	T4 (79–192 nmol/L)	TSH (0.27–4.2 mIU/L)			
1	21	2.15	<5.15	-	-	>100	8.3	56	2.8/2.9
2	20	1.76	6.63	1.76	30.80	>100	11.7	37	3.7/0.5
3	21	-	-	-	-	>100	11.1	-	1.8/10.0
4	22	3.65	5.13	1.78	46.75	>100	9.4	3(y)	3.5/7.3
5	27	-	-	0.31	-	>150	10.7	57	3.2/6.7
6	21	5.20	11.80	1.90	87.82	22.42	8.5	90	1.0/3.7
7	22	5.70	7.25	2.39	44.94	89.46	12.5	37	1.6/4.3
8	27	7.13	6.06	3.24	35.94	>100	9.4	46	2.0/0.8
9	17	2.89	<5.15	1.06	24.14	>100	10.7	31	1.9/0.9
10	20	4.89	<5.15	1.81	26.60	>100	15	34	1.9/2.0
11	36	6.33	9.01	-	-	68.54	4.9	97	1.0/8.5
12	17	6.96	10.17	-	-	46.41	6.8	60	0/4.5
13	17	1.62	2.96	-	-	>150	10.4	38	0/3.5
14	14	5.68	7.34	-	-	100	12.5	30	2.2/3.5
15	25	1.89	1.80	-	-	100	8.5	45	2.2/4.3
16	17	3.91	2.83	-	-	100	9.4	34	1.5/4.4
17	14	3.79	4.89	-	-	100	9.8	32	0.4/4.0
18	27	5.07	6.05	-	-	99.77	11.4	45	1.8/4.0
19	14	3.70	5.79	-	-	100	13.0	39	3.7/0.8
20	16	1.86	1.54	-	-	100	10.7	37	3.4/1.6
21	36	0.59	1.29	-	-	100	8.3	60	3.1/4.9
22	17	2.93	2.96	-	-	100	7.8	30	1.0/2.2
23	46	5.87	5.93	-	-	145.97	9.8	70	3.9/0.9
24	21	4.3	8.07	1.63	59.25	>100	7.4	71	3.3/2.5
25	90	-	-	-	-	>100	5.0	-	1.7/6.3
26	12	3.68	10.83	1.33	92.25	42.41	10	19	13.8/0.2
27	14	4.05	9.40	-	-	>150	7.0	90	2.1/8.3
28	22	-	4.03	1.1	-	223.1	7.6	50	1.6/8.3
29	15	4.56	11.84	1.41	57.27	64.89	12.5	21	0/0.1
30	13	3.70	6.69	-	-	100	13.0	25	1.5/3.7
31	14	1.39	2.19	-	-	100	13.0	45	0/4.3
32	11	1.48	2.19	-	-	100	8.6	120	4.7/2.9
33	18	2.19	2.71	-	-	56.5	7.7	30	3.1/2.1

T3: thyroxine 3; T4: thyroxine 4; FT3: free thyroxine 3; FT4: free thyroxine 4; TSH: thyroid-stimulating hormone; y: years; -: data lost or not tested.

## Data Availability

The datasets presented in this study are available upon request to corresponding author.

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
