# Peer review of "Genetic Factors Causing Thyroid Dyshormonogenesis as the Major Etiologies for Primary Congenital Hypothyroidism: Clinical and Genetic Characterization of 33 Patients"

_jcm, 2022, doi:10.3390/jcm11247313_

Round 1
Reviewer 1 Report
Reviewer comments on manuscript ID #jcm-2067325, entitled “Genetic factors causing thyroid dyshormonogenesis as the major etiologies for primary congenital hypothyroidism: Clinical and genetic characterization of 33 patients”,
Comment # 1, all manuscript: the authors should consider to carefully review the English language of the manuscript, preferably with a native English speaker. In the present form, the writing of this manuscript will likely distract the reader.
Comment # 2, Methods section: besides thyroid function tests and TSH cut-off to be selected for study, the authors do not provide any protocol for clinical assessment of patients. However, the authors present data on imaging in the results section. Thus, I do recommend the authors to describe in the Methods all tools that were used in the clinical assessment of patients with CH.
Comment # 3, Methods section: the authors do not provide any period of time where patients were selected, which is an important piece of information.
Comment # 4, Methods section: “Although sampling bias could not be ruled out considering the limited cohort size, these data strongly suggested that TDH rather than TD was the major mechanism for CH development.”. Worldwide, TD is recognized as the most common cause of CH. The authors included in their analysis only 33 patients and only 1 out of 8 genetic etiologies of TD (J Clin Endocrinol Metab. 2010;95(4):1981-5) were included. Thus, I would recommend the authors to at least describe what is published in the literature as the main cause of CH (TD) and respective genetic etiologies (genes).
Author Response
Dear Professor,
Thank you very much for managing and providing the opportunity for us to make substantial amendments on the manuscript (ID: jcm-2067325). The comments and the suggestions from the reviewers were really valuable for the authors to improve the quality of the manuscript. We have deliberated the comments/suggestions and made corrections accordingly. Attached below please find the point-to-point responses to the comments/suggestions in the letter.
With best regards.
Yuan-Zong Song, MD, PhD and professor,
Department of Pediatrics,
The 1st Affiliated Hospital,
Jinan University
Guangzhou, 510630
Guangdong,
China
Major comments
Comment 1, all manuscript: the authors should consider to carefully review the English language of the manuscript, preferably with a native English speaker. In the present form, the writing of this manuscript will likely distract the reader.
Response 1: This manuscript has undergone extensive English editing (English editing ID: 55315) by using one of the editing services listed at https://www.mdpi.com/authors/english. And the certification was attached.
Comment 2, Methods section: besides thyroid function tests and TSH cut-off to be selected for study, the authors do not provide any protocol for clinical assessment of patients. However, the authors present data on imaging in the results section. Thus, I do recommend the authors to describe in the Methods all tools that were used in the clinical assessment of patients with CH.
Response 2: Accordingly, we added a 2.2 Clinical evaluation section to describe in the Methods all tools including ultrasonography and Developmental quotient evaluation, in the revised version.
Comment 3, Methods section: the authors do not provide any period of time where patients were selected, which is an important piece of information.
Response 3: Added.
Comment 4, Methods section: “Although sampling bias could not be ruled out considering the limited cohort size, these data strongly suggested that TDH rather than TD was the major mechanism for CH development.”. Worldwide, TD is recognized as the most common cause of CH. The authors included in their analysis only 33 patients and only 1 out of 8 genetic etiologies of TD (J Clin Endocrinol Metab. 2010;95(4):1981-5) were included. Thus, I would recommend the authors to at least describe what is published in the literature as the main cause of CH (TD) and respective genetic etiologies (genes).
Response 4: To adress this issue, we added some published literatures and made relevant modifications on the Discussion section.

Reviewer 2 Report
Liu et al report on clinical and genetic characterization of 33 patients with primary congenital hypothyroidism.
The present MS is interesting, however it can not add relevant insights in this field in its present form. Criticism are in order.
1) PAX-8 variant in pat 22 was paternally derived, but no clinical information of the father are provided. Defective penetrance? If that was the case, it is very important to stress this aspect, to the benefit of proper genetic counselling to the familaies. In general, data on penetrance are recommended to be discussed through the whole MS with respect to the totality of causative genes identified so far.
2) the majority of the identified gene variants have AR inheritance. Segregation analysis in healthy siblings, if available, is very important to speculate about penetrance and possible oligogenic pathomechanisms
3) The discussion is very long and is to be significantly shortened.
4) Finally, a reader not closely involved in this field can not read the MS fluently in its present form.
Author Response
Dear Professor,
Thank you very much for managing and providing the opportunity for us to make substantial amendments on the manuscript (ID: jcm-2067325). The comments and the suggestions from the reviewers were really valuable for the authors to improve the quality of the manuscript. We have deliberated the comments/suggestions and made corrections accordingly. Attached below please find the point-to-point responses to the comments/suggestions in the letter.
With best regards.
Yuan-Zong Song, MD, PhD and professor,
Department of Pediatrics,
The 1st Affiliated Hospital,
Jinan University
Guangzhou, 510630
Guangdong,
China
Major comments
1) PAX-8 variant in patient 22 was paternally derived, but no clinical information of the father are provided. Defective penetrance? If that was the case, it is very important to stress this aspect, to the benefit of proper genetic counselling to the families. In general, data on penetrance are recommended to be discussed through the whole MS with respect to the totality of causative genes identified so far.
Response 1: Although detailed laboratory data was not available, the father of patient 22 had a history of hypothyroidism, as we added in Table 2. The penetrance was 100% in the 22 CH-affected children in this study, and we added this information in the 3rd paragraph from bottom of the Discussion section.
2) The majority of the identified gene variants have AR inheritance. Segregation analysis in healthy siblings, if available, is very important to speculate about penetrance and possible oligogenic pathomechanisms.
Response 2: The 33 CH patients in this study had no healthy siblings, but this suggestion on segregation analysis was significant and thus will be fully considered in our subsequent investigation on penetrance and possible oligogenic pathomechanisms. Thank you.
3) The discussion is very long and is to be significantly shortened.
Response 3: Done. The paragraphs 2 and 3 in the original version was shortened into a new paragraph 2 in the revised version.
4) Finally, a reader not closely involved in this field can not read the MS fluently in its present form.
Response 4: The manuscript has undergone extensive English editing to address this issue. And the certification was attached.
